# Freedom of Choice to Vaccinate and COVID-19 Vaccine Hesitancy in Italy

**DOI:** 10.3390/vaccines10111973

**Published:** 2022-11-21

**Authors:** Mawulorm K. I. Denu, Alberto Montrond, Rachael Piltch-Loeb, Marco Bonetti, Veronica Toffolutti, Marcia A. Testa, Elena Savoia

**Affiliations:** 1Emergency Preparedness Research Evaluation and Practice (EPREP) Program, Division of Policy Translation and Leadership Development, Harvard TH Chan School of Public Health, Boston, MA 02120, USA; 2Department of Biostatistics, Harvard TH Chan School of Public Health, Boston, MA 02120, USA; 3Carlo F. Dondena Research Center and COVID Crisis Lab, Bocconi University, 20136 Milan, Italy; 4Center for Evaluation Methods, Wolfson Institute of Population Health, Barts and The London School of Medicine and Dentistry, Queen Mary University of London, London E1 2AB, UK

**Keywords:** COVID-19, vaccine hesitancy, vaccine freedom

## Abstract

Despite the availability of effective vaccines that lower mortality and morbidity associated with COVID-19, many countries including Italy have adopted strict vaccination policies and mandates to increase the uptake of the COVID-19 vaccine. Such mandates have sparked debates on the freedom to choose whether or not to get vaccinated. In this study, we examined the people’s belief in vaccine choice as a predictor of willingness to get vaccinated among a sample of unvaccinated individuals in Italy. An online cross-sectional survey was conducted in Italy in May 2021. The survey collected data on respondents’ demographics and region of residence, socioeconomic factors, belief in the freedom to choose to be vaccinated or not, risk perception of contracting and transmitting the disease, previous vaccine refusal, opinion on adequacy of government measures to address the pandemic, experience in requesting and being denied government aid during the pandemic, and intent to accept COVID-19 vaccination. The analysis employed binary logistic regression models using a hierarchical model building approach to assess the association between intent to accept vaccination and belief in the freedom to choose to vaccinate, while adjusting for other variables of interest. 984 unvaccinated individuals were included in the study. Respondents who agreed that people should be free to decide whether or not to vaccinate with no restrictions on their personal life had 85% lower odds of vaccine acceptance (OR = 0.15; 95% CI, 0.09,0.23) after adjusting for demographic and socioeconomic factors and their risk perception of contracting and transmitting COVID-19. Belief in the freedom to choose whether or not to accept vaccinations was a major predictor of COVID-19 vaccine acceptance among a sample of unvaccinated individuals in Italy in May 2021. This understanding of how individuals prioritize personal freedoms and the perceived benefits and risks of vaccines, when making health care decisions can inform the development of public health outreach, educational programs, and messaging.

## 1. Introduction

The SARS-CoV-2 (COVID-19) pandemic has had a devastating effect on the world since the first case was detected in November 2019 [1]. It has resulted in millions of deaths worldwide [1,2] and caused the world economy several billions in losses [3,4]. Italy was the first European country to report clusters of cases in Cologno, a town near Milan in the northern region of Lombardy. Despite having a well-developed healthcare system [5], the country was quickly overwhelmed by the large and rapid surge in infections and deaths [6,7]. Early on in the pandemic, it became evident that non-pharmaceutical public health interventions alone were not enough, and vaccines were necessary. Globally, governments made huge investments in developing vaccines [8]. The COVID-19 vaccines were produced in record time thanks to the knowledge and expertise accrued in the past in the use of new technologies for vaccine production, significant investments in the production systems, and strategies to enhance efficiency in the execution of the experimental trials [9,10].

By November 2020, it was announced that the first vaccines had been developed with good efficacy against the severity of disease and mortality [11,12]. At that time, evidence on the impact of the vaccines on infectiousness were limited, the availability of which marked a major public health milestone in the pandemic. However, challenges were experienced in the supply chain [13,14] and allocation policies [15,16], and vaccine hesitancy in many countries.

Vaccine hesitancy is described as a delay or refusal in receiving a vaccination despite its availability [17,18]. Even before the pandemic, the World Health Organization (WHO) identified vaccine hesitancy as one of the top ten threats to global health [19]. Italy was one of the countries with the highest levels of hesitancy in the European Union [20,21]. Vaccine hesitancy is a complex issue, and understanding the factors driving it is important in developing interventions that address populations’ concerns about getting vaccinated [22].

Well-established demographic factors such as age, gender, race, fear of side effects, and untrusted efficacy and safety have been reported as main drivers of vaccine hesitancy in general [23,24,25,26,27,28,29,30]. The COVID-19 pandemic also rekindled other issues that affected vaccine acceptance including the long-standing debate around the freedom to choose whether to receive the vaccine or not. In a bid to increase vaccination rates, several countries and states introduced vaccine mandates and restrictions that were considered to infringe the personal liberties of citizens. Italy imposed one of the strictest mandate policies, which resulted in vivid political debates and civil protests [31,32,33]. Such policies included the COVID-19 certificates (“green pass”) strategy restricting access to public places for unvaccinated citizens or into the country for international visitors—unless able to produce proof of recovery from COVID-19 within the past 180 days or a negative COVID-19 test result [33,34]. Although it is unknown if this policy saved lives [35], it further intensified the global political debate about the freedom to vaccinate, with divided opinions [36,37,38].

In this study, we sought to examine the effect of the belief in the freedom to choose whether or not to accept vaccinations and its association with COVID-19 vaccine acceptance in Italy during the pandemic.

## 2. Materials and Methods

### 2.1. Data Collection

Data were collected using a cross-sectional study design with an online survey. The survey was implemented via mobile phones using the survey platform Pollfish [39] and was limited to individuals aged ≥ 18 across all regions in Italy. We aimed at a sample size of 1000 individuals based on a previous study [40] where approximately 20% of the population manifested hesitancy. This fraction was determined as sufficient to explore the association with socio-demographic and other factors. In addition, eligible participants in this study were identified using a screening question to limit the sample to those who were not vaccinated at all, and those who had received only one dose of a COVID-19 vaccine requiring two doses at the time of the survey. The Pollfish platform uses random device engagement (RDE), an approach like Random Device Dialing (RDD), to reach users engaged in using mobile applications (rather than calling them). Users are identified only by a unique device ID. Compared to RDD, RDE results in a higher response rate and avoids the potential bias of interviewer–respondent interactions. For this survey, a random sample of 1000 users who fit the study’s eligibility criteria was selected and data were collected between 21–28 May 2021. Only 16% of the Italian population had been fully vaccinated when the survey was launched [41].

All respondents were asked to consent to participate in the survey before proceeding to answer the survey questions. The study protocol and survey instrument were approved by the Harvard T.H. Chan School of Public Health Institutional Review Board (IRB) on 8 December 2020 (protocol #20-203) and were granted a checklist-based self-assessment exempt status at Bocconi University on 22 April 2021 (protocol #31146). Prior to implementation, feedback on the items was solicited from a small sample (*n* = 20) of individuals and used for item re-wording and incorporated in a revised version of the final questionnaire. The minimum length of time needed to thoughtfully complete the survey was estimated to be three minutes. That threshold was used as a criterion for data quality assurance, and any questionnaire completed in less than three minutes was removed from the analyses. The sample was equally distributed, using quotas by sex and age groups.

### 2.2. Dependent and Explanatory Variables

The dependent variable of interest was vaccine acceptance. This construct was measured in the survey by the question “If you were offered a COVID-19 vaccine at no cost, how likely are you to take it?” with response options (scoring): very likely (5), somewhat likely (4), I am not sure (3), somewhat unlikely (2), very unlikely (1) and I would not take it at the moment but would consider it later on (0). The responses were dichotomized into a binary variable coded as “Acceptant” if the answer was “very likely” and “Hesitant” for all other responses.

Table 1 presents the list of explanatory variables and how they were categorized.

The explanatory variable of main interest concerned the belief in the freedom to choose whether to receive the COVID-19 vaccine. This belief was assessed by measuring the individual’s agreement with the question: “People should be free to decide if they get vaccinated or not with no consequences for their job or personal life.” Responses were formatted using the 3-point Likert scale responses: “Disagree”, “Unsure”, and “Agree”.

Other variables included socio-demographics, risk perception (details below), history of vaccine hesitancy, and experience and opinions about the government response as described below.

#### 2.2.1. Socio-Demographics and Comorbidities

Data were collected on socio-demographic factors such as age, education, sex, citizenship, and region of residence. The region of residence was categorized into: “North” (Emilia-Romagna region and further north) and “South/Central (Tuscany and further south)”. The respondents experience with economic stress was measured by asking respondents “In the past 12 months (1 year) have there been occasions in which you were worried about not having enough money or resources to be able to have enough food to eat?” with responses coded as “No” (0) and “Yes” (1). Respondents were also asked to report if they were at high risk due to a health condition from a list of options of diseases or conditions associated with the risk of severity in case of COVID-19 infection such as diabetes, cancers, hypertension, and auto-immune diseases.

#### 2.2.2. Risk Perception

COVID-19 risk perception was measured using three questions that described perceived risk related to the spread of the virus: contracting COVID-19 at work, contracting COVID-19 outside of work, and infecting the subject’s family or friends. Respondents were asked to report their level of concern, namely not concerned (1), concerned (2), and very concerned (3) for each question. A principal component factor analysis was performed on the three items, and found that there was one main factor, and as a result the responses from each of the three questions were summed to create a summative three-item scale with values ranging from 3 to 9. Responses were then categorized into quintiles to create 5 levels of COVID-19 risk perception (1–5), where 1 represented the lowest level of risk perception and 5 represented the highest level.

#### 2.2.3. History of Vaccine Hesitancy

Previous vaccine hesitancy was measured using the question “In your life, were you ever recommended a vaccine (other than the COVID-19 vaccine) by a healthcare provider that you did not take?” to which respondents responded “Yes” or “No”.

#### 2.2.4. Experiences and Opinions Related to the Government Response to the Pandemic

Respondents were asked for their opinion on the accuracy with which the number of COVID-19 cases were reported to the public, and about the government transparency in providing information about the COVID-19 situation. Respondents were also asked about their experience in seeking government aid. Finally, they were asked how much they agreed with the statement “I think the measures taken so far by the government to respond to the COVID-19 pandemic has been:”. Specific questions and responses are provided in Table 2.

### 2.3. Statistical Analyses

We first produced descriptive statistics for each variable. To analyze how freedom of choice is associated with vaccine hesitancy, we performed a principal component factor analysis to explore the factor structure of the COVID-19 risk perception variable. Following this, we performed Chi-squared tests to examine differences in the dependent and independent variables between the two vaccine acceptance categories. The results are presented in Table 2, where column 2 presents the frequency for the vaccine acceptant group, column 3 for the vaccine hesitant group, and column 4 presents the *p*-value associated with the comparison. We then utilized three nested logistic regression models to examine the association between the predictors and COVID-19 vaccine acceptance with the following methodology: Model 1 estimated the association between vaccine acceptance and belief in the freedom to choose whether to accept vaccinations, and demographic variables (age, sex, region of residence). Model 2 included all the parameters from Model 1 and socioeconomic variables (educational level, employment status, and economic stress within the past 3 months). Model 3 included all parameters from Model 2 and previous vaccine hesitancy, COVID-19 risk perception, having experienced a rejection of a request for government aid during the pandemic, opinion of the accuracy of COVID-19 cases reported nationally, government transparency in providing information about the national COVID-19 situation, adequacy of government measures during the pandemic. The goodness-of-fit of the final model was tested using the Hosmer–Lemeshow test. The reported *p*-values are not adjusted for multiplicity, and all analyses were performed using STATA (Version 17).

## 3. Results

### 3.1. Socio-Demographic Characteristics of the Study Population

Table 1 presents summary statistics of the demographic characteristics of the study population. There were 984 respondents from all regions of Italy. 56.9% of respondents were very likely to accept the COVID vaccine, while 43.9% reported some hesitancy towards the vaccine.

### 3.2. Respondents’ Characteristics and Responses by Vaccine Acceptance

The first two columns of Table 2 summarize the comparisons of responses between the acceptant and hesitant groups of respondents by use of Chi-square tests on the (unordered) levels of the variables. The results highlight significant differences in all variables except for gender, region of residence, education, and employment status.

### 3.3. Results of Multivariate Logistic Regression Models

The last three columns of Table 2 provide a summary of the three multivariate regression analyses. In the most complete model (Model 3), respondents who agreed that people should be free to decide to get vaccinated or not against COVID-19 had 85% lower odds of accepting the vaccine, compared to participants who did not hold this belief (OR 0.15, 95% CI 0.09, 0.23). Those who were unsure had 63% decreased odds of accepting the vaccine (OR = 0.37, 95% C.I. 0.23–0.56) compared to those not holding the belief. Other variables such as region of residency, refusal of other vaccines in the past, and risk perception were associated with vaccine acceptance.

More specifically, respondents living in the Northern regions were more favorable about the vaccine with 40% increased odds of accepting the vaccination (OR = 1.4, 95% C.I. 1.04,1.89) compared to those living in the Central and Southern regions of the country. Those who refused other vaccines in the past had 61% decreased odds of accepting the vaccine (OR = 0.39, 95% C.I. 0.24–0.66) compared to those who did not.

Greater concern of contracting and transmitting COVID-19 was associated with 137% increased odds of vaccine acceptance (OR 2.37, 95% CI 1.44,3.90). This result is comparable to results from Model 1 and 2.

In addition, participants who requested government aid during the pandemic and were rejected had 38% decreased odds of accepting the COVID-19 vaccination (OR 0.62, 95%CI 0.39, 0.98) compared to participants who had not requested the aid and those who requested the aid and did not experience a rejection. Opinions about the appropriateness of the government response were also associated with vaccine acceptance. Those who believed the response was not right had 65% decreased odds of accepting the vaccine (OR = 0.35, 95% C.I. 0.26–0.48) compared to those who thought the response was just right. The association between the belief in freedom to choose whether or not to accept vaccinations and vaccine acceptance remained stable throughout all three models.

## 4. Discussion

The results of our study indicate that in Italy in May of 2021, when government officials were considering the implementation of a COVID-19 vaccine mandate as a measure to restrict movement and access to services for the unvaccinated, the belief in having the right to choose whether to get vaccinated or not was strongly associated with unwillingness of getting vaccinated among those who had not received the shot. The results of this study show that people who believed in the right to choose were less likely to accept the COVID-19 vaccine regardless of their age, sex, education, and risk perception of contracting and transmitting the disease [40,42]. It is also interesting to note that consistently with previous studies [43], only 1 out of 6 had refused vaccines in the past, and that the refusal to get the COVID-19 vaccine was specific to this disease for the majority of the respondents.

During the pandemic, people’s belief system was challenged abruptly, over a short period of time and without any preparation or education on the importance of accepting sudden changes—as a vaccine mandate—during emergency situations. It is possible that such policies are internalized by those more susceptible to vaccine hesitancy as a top-down order, which can be perceived, by some individuals, as a dictatorial move by the government trumping on individuals’ rights, liberties, and freedom in a free society. This becomes a point of contention and dispute over whether to accept or not accept the vaccine, given the drastic change and realities that develop quickly during a public health crisis. It can also be said that belief in personal freedom to vaccinate is underpinned by many factors including political orientation, religion, sociocultural beliefs, and other personal convictions that may explain hesitancy towards a policy imposed by the government [44,45,46,47,48]. These are lifelong beliefs that develop over an individual’s lifetime for multiple reasons, and can be considered as characterizing traits that are unlikely to change drastically even in the face of a deadly pandemic.

However, the causal pathway and directionality of whether the belief was formed or operationalized *after or before* the mandate or deliberations about the mandate went into effect or commenced cannot be determined from our cross-sectional design. If people did not want to get vaccinated initially because they considered themselves to be at low risk of the serious COVID-19 outcomes that the vaccine protected against, then discussion about a potential mandate could have internalized even more the challenge to their freedom to choose belief. In other words, our study cannot determine whether the belief was formed because of the contextual challenge to their assessment of their own personal risk regarding the vaccine, rather than an intrinsic trait of the individual. This “state versus trait” question is important for determining which predisposing factors should be considered when designing public health mitigation strategies since different approaches are more effective than others depending upon whether the belief is a “state” characteristic that is more amenable to change than a “trait” characteristic, unlikely to be influenced by a public health campaign.

Our findings also point to other important factors associated with vaccine hesitancy, including opinions regarding the government response to the pandemic and experience in seeking government support and seeing it be denied. In our sample, respondents who requested government aid during the pandemic and experienced a rejection of their request had lower odds of vaccine acceptance. It is possible that a rejection of financial aid might have undermined trust in the government’s ability to deliver services and protect its citizens, with unintended consequences on acceptance of the government mandate to vaccinate. This finding is important because it reflects the need for the government to provide clear information on who qualifies for aid and a realistic timeframe during which that aid can be delivered. Similarly, those who expressed a negative opinion about the government response to the pandemic were less likely to accept the vaccine. The findings of our study lead to some practical considerations for public communication efforts. First, we cannot assume that people will embrace rapid changes to their freedom of action with no preparation. Individuals need time to process information and rapid decisions undertaken during a crisis need an additional layer of explanations and support especially for those who are hesitant to act due to their personality. Second, governments should communicate the limitations of what the government can do during a pandemic and their sphere of action in order to reduce unrealistic expectations from the public. Third, any government action, including non-public health policies, has the potential to impact trust in institutions and trust in public health recommendations. Our study demonstrates the potential impact of denied government financial aid on vaccine acceptance. This result indicates the need for a whole of government approach during a crisis where clear communication should accompany the delivery of all policies to avoid misunderstanding and denial of services to those who believe to be entitled to receive them.

Finally, our findings suggest that vaccine hesitancy is the result of a complex decision-making process, entailing a variety of experiences and opinions that impact the individual overtime and can be strongly related to personal experiences in interacting with the government. All these factors should be taken into consideration when developing empathic and tailored communication strategies.

### Limitations

The use of a smartphone platform in sampling respondents was one of the limitations of this study. This convenient form of sampling may have introduced a sampling bias in our subject population [49]. Indeed, the older generation of respondents may not be well represented in this study population due to a higher tendency of having difficulties with using smartphones to answer surveys. The sample had 2% of non-citizens while the population of non-citizens in Italy is close to 10%. This could also be the result of the sampling method employed. Non-citizens, particularly undocumented non-citizens, may be less likely to respond to online or phone-based surveys.

Lastly, the cross-sectional design makes it difficult to establish a temporal sequence to infer causality between changes in beliefs and vaccine acceptance.

## 5. Conclusions

Belief in the freedom to choose whether or not to accept vaccinations was associated with greater COVID-19 vaccine hesitancy in Italy in May 2021, prior to the implementation of the vaccine mandate. Respondents who agreed with the statement that people should be free to decide to get vaccinated without consequences for their work or personal life had lower odds of vaccine acceptance compared to those who disagreed, regardless of their age and risk perception of contracting or transmitting the disease. In addition, the individual’s negative experience in seeking financial support from the government during the crisis and opinions related to the government response also impacted the willingness to get vaccinated.

## Figures and Tables

**Table 1 vaccines-10-01973-t001:** Demographic characteristics of study population.

(N = 984)	*n* (%)
Gender	
Male	489 (49.7%)
Female	495 (50.3%)
Age	
18–24	194 (19.7%)
25–34	195 (19.8%)
35–44	197 (20.0%)
45–54	198 (20.1%)
Over 54	200 (20.3%)
Education	
Less than high school	83 (8.4%)
High School of equivalent	440 (44.7%)
Some college	133 (13.5%)
Bachelor’s degree	277 (28.2%)
Post-graduate degree (ie Masters, Phd)	51 (5.2%)
Region of residence	
North	435 (44.2%)
Central/South	549 (55.8%)
Citizenship	
Italian Citizenship	961 (97.7%)
Non-Italian Citizenship	23 (2.3%)
Comorbidities	
None	563 (57.2%)
One or more	421 (42.8%)
Previous COVID-19 diagnosis	
No	885(89.9%)
Yes	99(10.1%)
Employment Status	
Employed	633 (64.3%)
Not employed	351 (35.7%)
Vaccine Acceptance	
Acceptant	560 (56.9%)
Hesitant	424 (43.1%)

**Table 2 vaccines-10-01973-t002:** Results of Chi-square analysis and logistic regression models. Outcome is vaccine acceptant (1) vs. hesitant (0).

	Acceptant	Hesitant	Model 1	Model 2	Model 3
Variables	(*n* = 560)	(*n* = 424)	OR	OR	OR
People should be free to decide if they get vaccinated or not with no consequences for their job or personal life					
Disagree	168 (30.0%)	149 (26.6%)	Ref	Ref	Ref
Unsure	243 (43.4%)	156 (36.8%)	0.31 ***	0.32 ***	0.37 ***
			(0.20,0.47)	(0.21,0.50)	(0.23,0.56)
Agree	149 (26.6%)	156 (36.8%)	0.12 ***	0.13 ***	0.15 ***
*p*-value (χ^2^)	<0.001		(0.08,0.19)	(0.08,0.20)	(0.09,0.23)
Age					
18–24	116 (20.74%)	78 (18.4%)	Ref	Ref	Ref
25–34	118 (21.1%)	77 (18.2%)	1.06	1.11	1.1
			(0.68,1.63)	0.71,1.74)	(0.68,1.77)
35–44	100 (17.9%)	97 (22.9%)	0.73	0.75	0.65
			(0.48,1.12)	(0.48,1.17)	(0.40,1.04)
45–54	100 (17.9%)	98 (23.1%)	0.82	0.87	0.72
			(0.53,1.25)	(0.56,1.36)	(0.44,1.16)
55+	126 (22.5%)	74 (17.5%)	1.36	1.39	1.2
			(0.88,2.11)	(0.88,2.17)	(0.73,1.95)
*p*-value (χ^2^)	0.026				
Sex					
Male	273 (48.8%)	216 (50.9%)	Ref	Ref	Ref
Female	287 (51.3%)	208 (49.2%)	1.15	1.16	1.08
			(0.87,1.52)	(0.87,1.55)	(0.79,1.47)
*p*-value (χ^2^)	0.496				
Region of Residence					
Southern/Central	257 (45.9%)	178 (42.0%)	Ref	Ref	Ref
North	303 (54.1%)	246 (58.0%)	1.19	1.17	1.40 *
			(0.91,1.57)	(0.89,1.55)	(1.04,1.89)
*p*-value (χ^2^)	0.221				
Education					
Less than high school	45 (8.0%)	38 (9.0%)		Ref	Ref
High school or equivalent	236 (42.1%)	204 (48.1%)		0.97	0.88
				(0.58,1.62)	(0.52,1.51)
Some college	86 (15.4%)	47 (11.1%)		1.28	1.22
				(0.69,2.35)	(0.64,2.31)
Bachelor’s degree					
	167 (29.8%)	110 (25.9%)		1.13	0.94
				(0.66,1.95)	(0.53,1.67)
Postgraduate degree (Masters, MD, PhD)	26 (4.6%)	25 (5.9%)		0.82	0.72
				(0.38,1.77)	(0.32,1.62)
*p*-value (χ^2^)	0.114				
Employment Status					
Unemployed	207 (37.0%)	144 (34.0%)		Ref	Ref
Employed	353 (63.0%)	280 (66.0%)		0.9	1.01
				(0.66,1.23)	(0.72,1.42)
*p*-value (χ^2^)	0.33				
Economic Stress in past 3 months					
No	390 (69.6%)	262 (61.8%)		Ref	Ref
Yes	170 (30.4%)	162 (38.2%)		0.73 *	0.77
				(0.54,0.98)	(0.55,1.07)
*p*-value (χ^2^)	0.01				
Request for Government Aid					
No aid rejected	513 (91.6%)	362 (85.4%)			Ref
At least one request rejected	47 (8.4%)	62 (14.6%)			0.60 *
					(0.38,0.96)
*p*-value (χ^2^)	0.002				
Previous Hesitancy to other vaccines					
No	33 (5.9%)	59 (13.9%)			Ref
Yes	527 (94.1%)	365 (86.1%)			0.39 **
					(0.24,0.66)
*p*-value (χ^2^)	<0.001				
COVID-19 Risk Perception Concern Quintile					
1-Lowest concern quintile	85 (15.2%)	93 (21.9%)			Ref
2	108 (19.3%)	106 (3%)			0.99
					(0.62,1.57)
3	151 (27.0%)	97 (22.9%)			1.42
					(0.90,2.26)
4	95 (17.0%)				1.83 *
		58 (13.7%)			(1.08,3.11)
5-Highest concern quintile	121 (21.6%)	70 (16.5%)			2.14 **
					(1.30,3.54)
*p*-value (χ^2^)	0.003				
Opinion on accuracy of COVID-19 cases reported					
Somewhat accurate	257 (45.9%)	140 (33.0%)			Ref
Not accurate	303 (54.1%)	284 (66.9%)			0.83
					(0.61,1.13)
*p*-value (χ^2^)	<0.001				
Opinion on government transparency in providing information about the national COVID-19 situation					
Government has been transparent	453 (80.9%)	289 (68.2%)			Ref
Government has not been transparent	107 (19.1%)	135 (31.8%)			0.88
					(0.62,1.26)
*p*-value (χ^2^)	<0.001				
I think that most of the measures taken so far by the government to respond to the COVID-19 pandemic were:					
Right	335 (59.82%)	133 (31.37%)			Ref
Not right	225 (40.18%)	291 (68.63%)			0.35 **
					(0.26,0.48)
*p*-value (χ^2^)	<0.001				

Key * = *p*-value < 0.05. ** = *p*-value < 0.01. *** = *p*-value < 0.001. Model 1—Adjusted for age, sex and geographical location. Model 2—Adjusted for age, sex, geographical location, education status and economic stress within the past 3 months. Model 3—Adjusted for age, sex, geographical location, education status, economic stress within the past 3 months, request for government aid, previous hesitancy to other vaccines, COVID-19 concern quintiles, opinion on accuracy of COVID-19 cases reported, opinion on government transparency in communicating about the national COVID-19 situation, and opinion on adequacy of government measures in responding to the pandemic.

## Data Availability

Data will be made available upon reasonable request.

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
