# Peer review of "Freedom of Choice to Vaccinate and COVID-19 Vaccine Hesitancy in Italy"

_vaccines, 2022, doi:10.3390/vaccines10111973_

Round 1

Reviewer 1 Report

The authors of the present manuscript tried to investigate whether the hesitancy of vaccinating for the COVID-19 vaccine in Italy was correlated with the mandatory strategy adopted by the Italian government. The study is of worth publishing.

A general comment is that the authors did not consider/study the correlation of the hesitancy for vaccination with this certain vaccine with the novel technology of the vaccine, that several scientists proposed as not safe for mass vaccination. 

Lines 47-49 and Line 52: please rephrase in terms of better English language. It is not clear what authors want to say

Lines 51-52: "it was announced that the first vaccines had been developed with good efficacy against the severity of disease and mortality". That time, November 2020 "immunity wall" was also one of the main arguments for convincing of mass vaccination, which was very soon confuted. 

Line 73: there are not many studies supporting that "green pass" saved lives, even Ref 34 states in the abstract that "although causality cannot be directly inferred".  There is not sufficient justification for this statement, that "green pass" saved lives.

Author Response

Lines 47-49 and Line 52: please rephrase in terms of better English language. It is not clear what authors want to say.

The text has been revised for clarity.

Lines 51-52: "it was announced that the first vaccines had been developed with good efficacy against the severity of disease and mortality". That time, November 2020 "immunity wall" was also one of the main arguments for convincing of mass vaccination, which was very soon confuted. 

A sentence was added.

Line 73: there are not many studies supporting that "green pass" saved lives, even Ref 34 states in the abstract that "although causality cannot be directly inferred".  There is not sufficient justification for this statement, that "green pass" saved lives.

Sentence modified to acknowledge this.

Reviewer 2 Report

This paper applies excellent statistical models for the analyses of the survey data and follow standard statistical protocols for the analysis. The analyses appear to be technically sound. The one suggestion that I have for the authors is that it would be very good if you find any additional insights in your analyses for how vaccine acceptance could be encouraged and improved. 

Author Response

This paper applies excellent statistical models for the analyses of the survey data and follow standard statistical protocols for the analysis. The analyses appear to be technically sound. The one suggestion that I have for the authors is that it would be very good if you find any additional insights in your analyses for how vaccine acceptance could be encouraged and improved. 

Additional insights on the implications of the findings and recommendations are now provided in the discussion.

Reviewer 3 Report

This is a cross-sectional study focusing on the freedom of a person take or refuse the Covid-19 vaccine against in Italy. Data on demography and other factors were collected and analyzed using robust statistical analysis. In addition, the authors nicely have stated the limitation of the study. However, a few points are not clear, particularly the way the results are presented. Comments are as follows:

In methodology the description of models 1,2,and 3 is not clear, can these be presented in a Table or graphically for better understanding, please?

Sample size 984, how do you sure that this number is enough for the claimed findings of this study…show a sample size calculation. Or some sort of justification to justify that 984 sample size is ok or enough for this type of study??

Line 125-126, why font size bigger??

Line 131, what were in that list of options of diseases...please mention it

Line 237:  any explanation why in the past only 1 out of 6 had refused vaccines ??

Discussion section needs  to be improved

Author Response

In methodology the description of models 1,2,and 3 is not clear, can these be presented in a Table or graphically for better understanding, please?

Table with description of models and component variables will be provided in supplementary materials.

Sample size 984, how do you sure that this number is enough for the claimed findings of this study…show a sample size calculation. Or some sort of justification to justify that 984 sample size is ok or enough for this type of study??

The size was determined based on a previous study. This process is now cited in the methods.

Line 125-126, why font size bigger??

This error has been corrected.

Line 131, what were in that list of options of diseases...please mention it.

Diseases mentioned.

Line 237:  any explanation why in the past only 1 out of 6 had refused vaccines ??

This rate is consistent with rates of vaccine hesitancy reported in Italy before the COVID-19 vaccination campaign.

Reference: Alessandro Sindoni.; Valentina Baccolini.; Giovanna Adamo. et al. Effect of the mandatory vaccination law on measles and rubella incidence and vaccination coverage in Italy (2013-2019). Human Vaccines & Immunotherapeutics. 2022; 18:1. DOI: 10.1080/21645515.2021.1950505

Discussion section needs  to be improved

The discussion has been improved by describing 3 key implications of the study findings.